# Catalytic Dehydration of Isopropanol to Propylene

Jean-Luc Dubois [1,*], Georgeta Postole [2], Lishil Silvester [2] and Aline Auroux [2]

[1]  Arkema France, Corporate R&D, 420 Rue d'Estienne d'Orves, 92705 Colombes, France
[2]  Univ Lyon, Université Claude Bernard Lyon 1, CNRS, IRCELYON, F-69626 Villeurbanne, France
*  Correspondence: jean-luc.dubois@arkema.com; Tel.: +33-472398511

**Abstract:** Catalytic dehydration of isopropanol to propylene is a common reaction in laboratories to characterize the acid–base properties of catalysts. When acetone is produced, it is the sign of the presence of basic active sites, while propylene is produced on the acid sites. About 2/3rd of the world production of isopropanol is made from propylene, and the other third is made from acetone hydrogenation. Since the surplus acetone available on the market is mainly a coproduct of phenol synthesis, variations in the demand for phenol affect the supply position of acetone and vice versa. High propylene price and low demand for acetone should revive the industrial interest in acetone conversion. In addition, there is an increasing interest in the production of acetone and isopropanol from $CO/CO_2$ via expected more environmentally friendly biochemical conversion routes. To preserve phenol process economics, surplus acetone should be recycled to propylene via the acetone hydrogenation and isopropanol dehydration routes. Some critical impurities present in petrochemical propylene are avoided in the recycling process. In this review, the selection criteria for the isopropanol dehydration catalysts at commercial scale are derived from the patent literature and analyzed with academic literature. The choice of the process conditions, such as pressure, temperature and gas velocity, and the catalysts' properties such as pore size and acid–base behavior, are critical factors influencing the purity of propylene. Dehydration of isopropanol under pressure facilitates the downstream separation of products and the isolation of propylene to yield a high-purity "polymer grade". However, it requires to operate at a higher temperature, which is a challenge for the catalyst's lifetime; whereas operation at near atmospheric pressure, and eventually in a diluted stream, is relevant for applications that would tolerate a lower propylene purity (chemical grade).

**Keywords:** isopropanol; propylene; acetone; diisopropyl ether; dehydration; process

## 1. Introduction

Propylene's main application is in the synthesis of polypropylene, but it is also used in numerous applications in the chemical industry such as the synthesis of acrylonitrile, acrylic acid, propylene oxide, propylene glycol, etc., but also for phenol synthesis. In the latter case, propylene is reacted with benzene to produce cumene. Cumene is further oxidized to phenol and acetone. Since the markets of these two products are completely different, their prices can vary in opposition, which compromises plants' expansions. Acetone hydrogenation to isopropanol/isopropyl alcohol (IPA), followed by its dehydration to propylene allows the loop to close and to be independent of acetone market fluctuations. According to IHS, such a process was implemented in Japan by Mitsui, which had several related patents [1]. More recently, Mitsui communicated on its acetone hydrogenation process to synthesize IPA [2] and on its biobased IPA process to synthesize propylene [3].

Besides this process, there are currently numerous ongoing research activities aiming to develop new synthetic routes to acetone and/or IPA. For example, acetone was produced during the First World War by the Acetone–Butanol–Ethanol (ABE) fermentation process. Acetone was an important chemical compound for the production of a smoke-free explosive. Whiskey distilleries were converted to produce this important chemical, but the plants and processes did not survive the boom of the petrochemical industry in the Western world.

More recently, several companies have tried to reintroduce the process to the market but struggled to achieve an acceptable price premium. This process also suffers from the fact that it also generates a major coproduct (*n*-butanol) which also has to find a market.

Another process to produce acetone is the reaction of two acetic acid molecules, releasing acetone and $CO_2$, known as the calcium acetate dry distillation. This process has been extended to longer chain volatile fatty acids (VFA), which include also propionic, butyric, isobutyric, valeric, and hexanoic acids, in order to obtain a mix of ketones which can be further converted to light fuels including gasoline. This was the technology that the US start-up Terrabon intended to implement to make a green gasoline.

There have also been some attempts in the past to produce acetone via the indirect oxidation of propane [4]. Acetone is a common solvent in the chemical industry and a valuable source of more sustainable acetone is to use recycled solvent. Indeed, acetone can be easily purified by distillation because of its low boiling point. The volumes on the market are limited, but the prices can be attractive [5].

IPA can also be produced by fermentation to Isopropanol–Butanol–Ethanol (IBE), through a modified version of the ABE process. It is otherwise produced by direct hydration of propylene or by hydrogenation of acetone [6,7].

Besides these routes from renewable resources and recycling, acetone has recently attracted some interest for production from CO or $CO_2$ through biochemical conversion [8,9]. There are still several challenges to overcome in this route, and derivatives are worth investigating.

Acetone recycling to propylene might be unavoidable in the future, as the gap in demand for phenol and acetone widens, while nearly 97% of the acetone on the market today is a coproduct of phenol synthesis. Moreover, some large consumers of acetone for methyl methacrylate (MMA) might shift their production to new processes using ethylene instead of acetone as feedstock (the ALPHA process from Lucite/Mitsubishi, and the LIMA process from Evonik/Roehm), leaving a significant volume of acetone on the market [10,11]. In addition, PMMA is a polymer which is easy to recycle through depolymerization, giving a high yield and high purity to MMA and a low carbon footprint, with appropriate technologies [12].

## 2. Process Conditions

### 2.1. Acetone, Isopropanol and Propylene Impurities

Depending on the process route, acetone and IPA contain different impurities that have to be taken into account. Propylene is also marketed with different grades which depend on the foreseen application (e.g., Polymer Grade or Chemical Grade), see Table 1.

Acetone made by the classical cumene oxidation process will always contain some trace amounts of aromatics, especially benzene, which has a boiling point close to acetone. Acetone produced by fermentation, however, would have no (or very low) traces of aromatics, and this would be a key technical advantage in some applications such as cosmetics, for example. Recycled acetone could contain impurities depending on the process in which the solvent was used primarily. Acetone obtained from acetic acid and VFAs could be contaminated by other ketones, most likely methylethylketone (MEK).

IPA produced by hydration of propylene would not make much sense if the goal was to dehydrate it back to propylene, except in the cases where propylene is mixed with hydrocarbons, and hydration would be seen as a way to extract the olefins as alcohols.

Acetone and IPA can be further purified, but this will come with an extra cost. Depending on the final application, this might not be necessary. For example, when acetone is hydrogenated to IPA, any poisons for the hydrogenation catalyst should be removed. Similarly, when IPA is dehydrated to propylene, it will generate water and some side products. Water cannot be considered as a severe poison for the dehydration reaction; however, some coke precursors can deactivate the catalysts.

**Table 1.** Polymer grade and chemical grade propylene. Example of compositions.

| Product | High Purity–Polymer Grade | | Chemical Grade | Methods |
| | Molar Basis | Weight Basis | Molar or Volume Basis | |
|---|---|---|---|---|
| CO | 1.0 ppm mol max | 0.67 ppm wt max | 1 ppm mol max | ASTM D-2504 |
| $CO_2$ | 1.0 ppm mol max | 1.0 ppm wt max | 15 ppm mol max | ASTM D-2505 MA0743-UP050 |
| COS | 0.02 ppm mol max | 0.03 ppm wt max | — | ASTM D-5303 |
| $H_2O$ | 12.0 ppm mol max | 5.0 ppm wt max | 10 ppm mol max | Online analyzer MA0743-UP051 |
| Hydrogen | 20 ppm mol max | 1 ppm wt max | 20 ppm mol max | ASTM D-2504 |
| Oxygen | 3.0 ppm mol max | 2.0 ppm wt max | 5 ppm mol max | Online analyzer SRPS H.B8.700 (2015) ASTM 2504-88 (2015) |
| Nitrogen | — | — | 50 ppm mol max | SRPS H.B8.700 (2015) ASTM 2504-88 (2015) |
| Methanol | 5.0 ppm mol max | 4.0 ppm wt max | — | By GC |
| Ammonia | 0.50 ppm mol max | 0.20 ppm wt max | — | — |
| Phosphine | 0.10 ppm mol max | 0.10 ppm wt max | — | Rateometric colorimetry or GC mass spectrometry |
| Arsine | 0.01 ppm mol max | 0.01 ppm wt max | — | — |
| Methane + Ethane | 500 ppm mol max | 357 ppm wt max | 1000 ppm mol max | ASTM D-2712 |
| Methyl Acetylene | 2.0 ppm mol max | 2.0 ppm wt max | 10 ppm mol max | ASTM D-2712 |
| Acetylene | 2.0 ppm mol max | 1.0 ppm wt max | 5 ppm mol max | ASTM D-2712 |
| Ethylene | 15 ppm mol max | 10 ppm wt max | 500 ppm mol max | ASTM D-2712 |
| Propadiene | 1.0 ppm mol max | 1.0 ppm wt max | 30 ppm mol max | ASTM D-2712 |
| Butadienes | 1.0 ppm mol max | 1.3 ppm wt max | 50 ppm mol max | ASTM D-2712 |
| C4′s | 10 ppm mol max | 13 ppm wt max | 2000 ppm mol max | ASTM D-2712 |
| Propane | 5000 ppm mol max | 5241 ppm wt max | 7.0% mol max | ASTM D- 2712 |
| Propylene | 99.50 Mole % min | 99.48 wt.% min | 93.0% mol min | ASTM D-2712 |
| Total Sulfur | 1.3 ppm mol max | 1.0 ppm wt max | 10 ppm wt max | ASTM D-4045 SRPS B.H8. 125 (2015) |

**Sources:** Composition based on datasheets from Chevron Philipps, high-purity propylene MSDS #5349 (2006) for polymer grade and from PetroHeija for chemical grade propylene composition. Note: C4s include iso-butane, *n*-butane, 1-butene, t-2-butene, c-2-butene, and iso-butene.

## 2.2. Purification Challenges

IPA dehydration is done with liquid or solid acid catalysts, but in view of the high corrosion risks and expensive materials required for liquid catalysts, a heterogeneous catalyst is largely preferred.

A challenge in IPA dehydration is then the separation of the reaction products. IPA dehydration leads to an increase in the number of moles, so the reaction is not favored by an increase in pressure. As all dehydration and hydration reactions are equilibrium limited, the reverse reactions should also be considered. However, water can also impact the side reactions.

Water and IPA form an azeotrope (31.67 mol% water) which make their separation more difficult [13], and it would be relevant to accept some water in the IPA stream anyways. If the IPA conversion in not complete in the dehydration reactor, the recycling of the IPA–water azeotrope would have to be considered, and in that case, some water would be fed to the dehydration reactor regardless.

Operating under pressure is not favored by thermodynamics when considering merely the reaction. However, when looking at the downstream purification it would have several advantages. The reaction products at full conversion will be water and propylene. There might be some trace amounts of remaining IPA because the reaction is equilibrium limited. When the stream is cooled, propylene remains in the gas phase while the water and IPA

traces are in the liquid phase. If the reaction is done at atmospheric pressure, there will be always some water partial pressure in propylene (Figure 1).

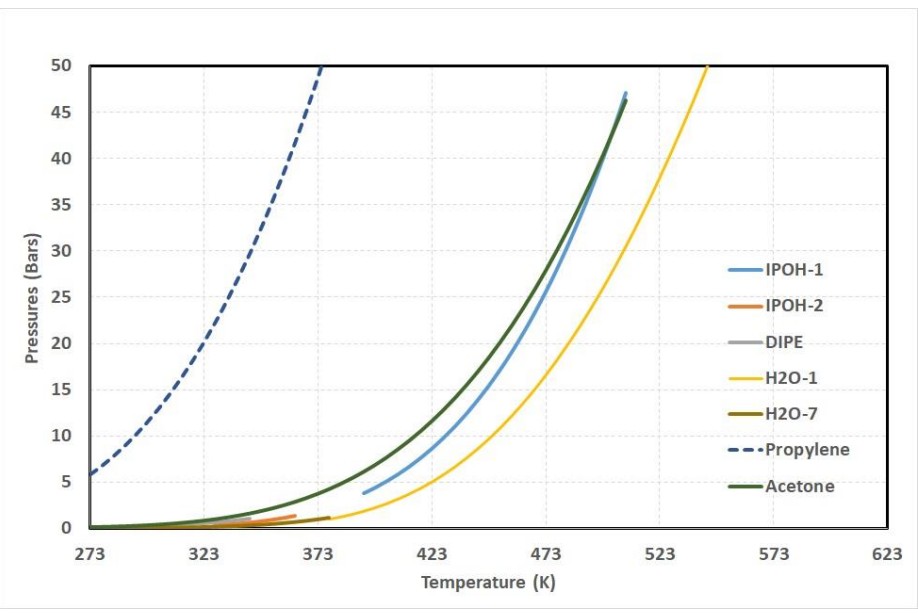

**Figure 1.** Figure built on Antoine equation parameters available from the NIST Webbook for isopropanol (IPOH-1 and IPOH-2), diisopropylether (DIPE), water/steam ($H_2O$-1 and $H_2O$-2), propylene and acetone.

### 2.3. Impurities Generated during the Dehydration Reaction

IPA can also dehydrate into diisopropylether (DIPE). On some basic sites, it would dehydrogenate back to acetone. Additionally, propylene can be rehydrated to n-propanol, which in turn can be dehydrogenated to propanaldehyde.

The choice of the operating conditions also depends on the end-use applications. If chemical grade propylene is to be used for acrylic acid production by selective oxidation, the presence of trace amounts of water is not a problem since usual process conditions require some water in the reaction gas regardless. However, propanaldehyde content should be extremely low.

In other applications, the presence of water has to be limited. For example, in acrylonitrile synthesis, although water is a reaction product, it is not present in the reagent stream. For polypropylene application, "polymer grade" purity must be reached.

### 2.4. Impact of Operating Pressure

When carrying out the dehydration process at 300 °C (573 K) and 30 bars pressure, all reagents and products are in the gas phase (see Figure 1). After reaction, when the temperature is reduced below 150 °C (423 K), propylene is still in the gas phase while the other products are mostly in the liquid phase and can be separated. When the gas is cooled further, propylene will eventually condense and can be easily stored and transported. In addition, propylene can be further purified when re-evaporated/distilled.

These thermodynamic constraints already dictate some process conditions and key properties for the catalysts. We can then consider two major cases:

- a dehydration under pressure (e.g., 30 bars), where a high purity of propylene could be achieved;
- a dehydration at or close to atmospheric pressure in which some water and IPA could remain in small quantities.

## 2.5. Impact of Operating Temperature

When the reaction is carried out under pressure and in the gas phase, it should be operated above 200 °C, preferably above 250 °C; while for dehydration at atmospheric pressure, a temperature above 100 °C might be sufficient. In the first case, catalysts have to be inorganic oxides, and they have to sustain some partial pressure of steam (steaming can severely damage catalysts over time). In the second case, polymeric acidic catalysts can be considered, such as acidic resins. However, the polymeric nature of the catalyst will limit the operating temperature, and the equilibrium between all reactions might limit the conversion.

If the endothermic dehydration reaction is carried out at too high of a temperature, over-cracking and polymerization could occur that can not only reduce the yield but also generate impurities that are difficult to separate.

## 2.6. Diisopropylether Conditions

DIPE is mainly produced by the dehydration of IPA, but it can also be produced by the reaction of IPA on propylene (alcoholysis reaction). It is part of the equilibrium with IPA, water, and propylene. To shift the equilibrium, higher temperatures are preferable. In most cases where DIPE formation is reported, the temperature is below 200 °C.

## 2.7. Isopropanol and DIPE Safety Concerns

Besides being flammable, IPA and DIPE are recognized as peroxide forming solvents Group B [14]. Several accidents have been reported in laboratories due to long-standing open flasks exposed to light. Therefore, it is recommended to check the IPA for peroxide content on a regular basis to avoid the risk of explosion when it is evaporated. The authors checked the French database of industrial accidents (which also includes major accidents abroad) for IPA and DIPE cases [15]. Out of 33 IPA and three DIPE cases, only one was related to an explosion during a distillation operation (but the causes are not fully explained). In most cases, IPA was only present on the site but not directly involved, and a large number of cases are related to transportation and storage.

Of course, if the reaction is operated under pressure at the laboratory, glass reactors are replaced with pressurized flow-through reactors, which are not much different from usual hydrogenation reactors.

## 3. Isopropanol Dehydration as a Test Reaction

### 3.1. Solid Acid–Base Test Reaction

IPA dehydration is a well-known test reaction in catalyst screening employed to evaluate the acid–base properties of catalysts. Acidic catalysts will catalyze dehydration, while basic catalysts will favor the dehydrogenation to acetone. This test reaction is relevant at low conversion rates when no product formations are limited by equilibrium. It also means that at low conversion rates, the amount of water produced is low. In addition, the reaction is often done at a low partial pressure of IPA (IPA is diluted in an inert gas), low total pressure, and low contact times (see Table 2). So, the preferred reaction conditions are not the same as for a commercial process which targets large production capacities.

### 3.2. Reaction Mechanism

Decomposition (dehydration and dehydrogenation) of IPA can proceed through different mechanisms depending on the acid–base nature of the catalysts. It is widely accepted that propylene is formed by dehydration of IPA via E1, E1cb, or E2 mechanisms [16–18] (Figure 2). The E1 mechanism is a two-step mechanism involving the cleavage of the C-OH bond resulting in an intermediate carbenium ion that undergoes subsequent proton loss to form propylene [17–20]. The E1 mechanism generally requires acid sites and can operate without any basic sites. E1cb is also a two-step process similar to E1, but an isopropyl carbanion intermediate is formed as the β-hydrogen elimination takes place first followed by the cleavage of the C-OH bond to produce propylene [18,19,21]. In addition to the

acid sites, E1cb also requires the basic sites for hydrogen abstraction (carbanion formation) whereas in the formation of propylene by the one-step E2 mechanism, the concerted water elimination occurs simultaneously by the breaking of the C-OH and β-H bonds of an IPA molecule that involves both the acid and basic sites [18,19,21].

DIPE is generated by inter-molecular dehydration via the SN1 or SN2 mechanism [17,22,23]. The SN1 mechanism is a two-step mechanism in which an isopropoxide carbanion (nucleophile) which is formed by β-hydrogen abstraction over the basic sites subsequently replaces the hydroxyl group of another IPA molecule to form DIPE. The acid sites can generally facilitate the replacement of the hydroxyl group by partially binding the hydroxyl groups onto the acid sites. If the aforementioned nucleophilic attack and hydroxyl group removal occur simultaneously to generate an ether, then it is an SN2 mechanism. Hence, both acid and basic sites in the catalysts are normally involved in DIPE production. It is also possible that the etherification of IPA and propylene takes place to form DIPE at higher conversion rates [24].

Acetone is the IPA dehydrogenation product formed via E1cb mechanism, i.e., the two-step process [16,21]. Generally, the isopropoxide species formed over basic sites undergo α-hydrogen abstraction forming a carbanion intermediate that finally leads to acetone. Acetone can also be formed by dehydrogenation on metallic catalysts containing redox sites and particularly in presence of oxidizing atmosphere via the Mars–van Krevelen mechanism [16,25].

As soon as the conversion and IPA partial pressure increase, the amount of water formed increases. This means that any Lewis site on the catalyst is likely to be converted to a Brønsted site. A working catalyst in a commercial process, i.e., long reaction tubes, could have completely different type of active sites between the entrance and the exit of the reactor. In addition, these sites can be deactivated by carbon deposition and other deactivation mechanisms such as surface reconstruction. Unfortunately, catalysts after initial activation have been rarely analyzed in as much detail as the fresh catalysts.

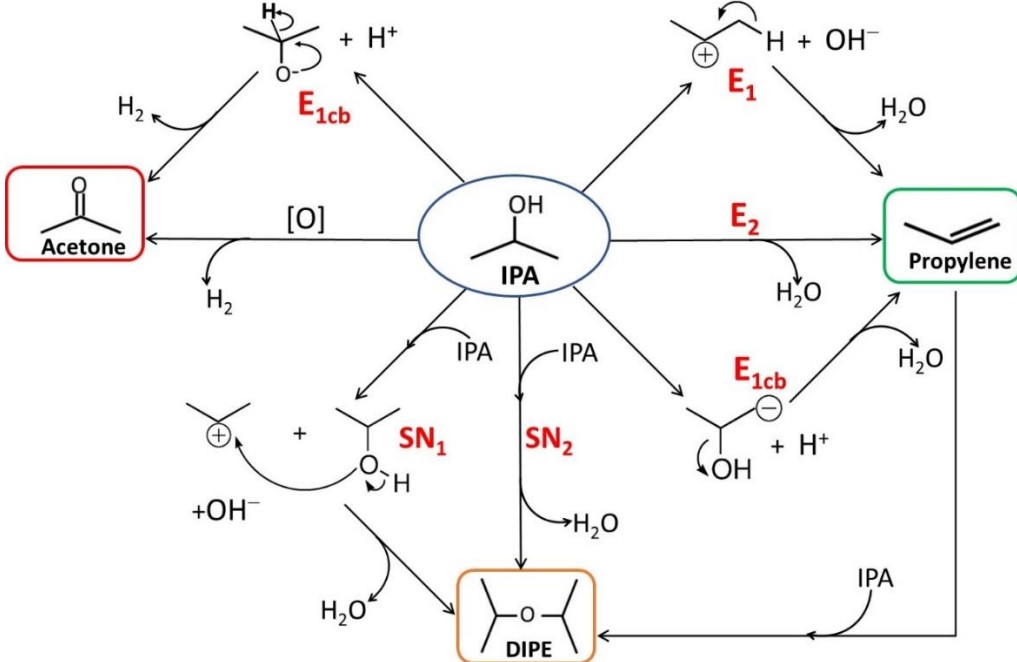

**Figure 2.** Schematic representation of the possible isopropanol (IPA) reaction mechanisms (adapted from [17,24]).

### 3.3. Catalysts Investigated

In the 1980s, almost half of the publications were on aluminas, silicas, silica–aluminas, and zeolites, while they currently represent only about 30%. Formulations containing Zirconium, Nobium, and Gallium have emerged (Figure 3).

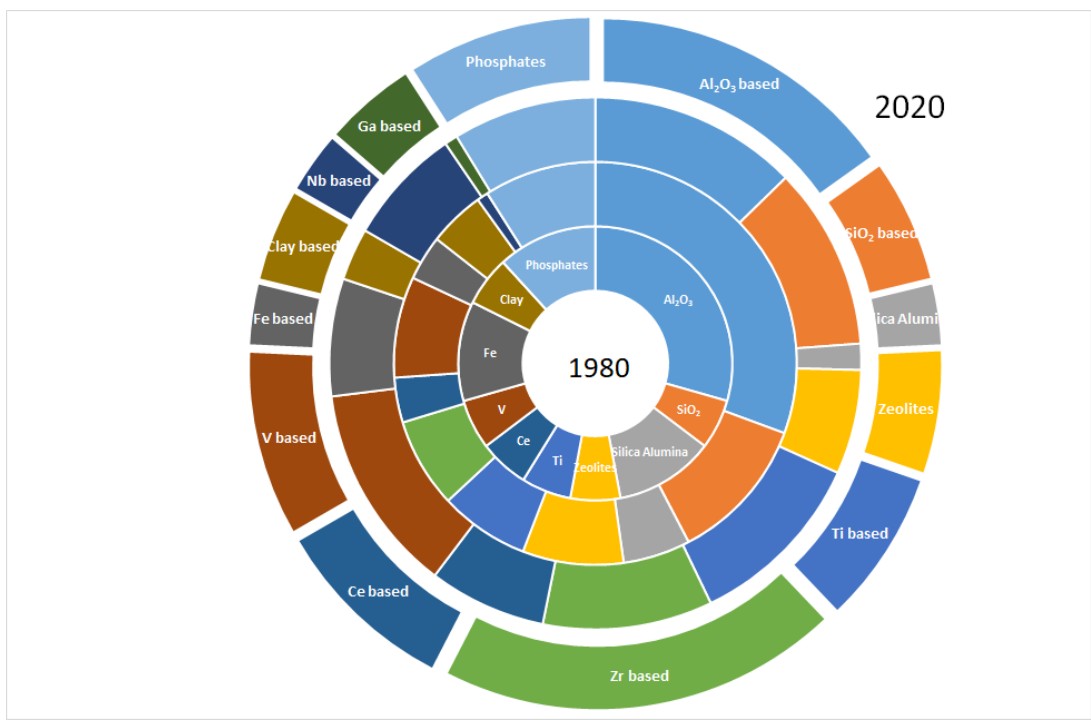

**Figure 3.** Evolution of the catalyst families that have been reported during the last 4 decades (from 1980 till 2020). Data collected from web of science.

Several decades ago, the catalysts investigated for IPA dehydration reaction were simple or mixed oxides, zeolites, and other acidic catalysts. Recent exotic catalysts have a low probability to be implemented because (i) they either contain Critical Raw Materials [26] such as noble metals, tungsten, boron, phosphorous, vanadium, gallium, etc., or (ii) they are produced with chemicals that might be listed as Substances of Very High Concern (SVHC) due to their toxicity [27] or in the SINLIST [28], which anticipates the chemicals that may end up in the SHVC list.

In Table 2, some catalysts which have shown high IPA conversion rates and high propylene selectivity are reported. The active sites mentioned in Table 2 are the possible sites that can be advantageous for the production of propylene. Additionally, it is evident that these sites can vary with the catalysts and experimental conditions. Moreover, the LAS and BAS reported here for each catalyst are the sites present in the fresh catalysts, which are prone to undergo considerable changes during the dehydration reaction at higher temperatures, especially in the presence of water. For all these examples in which tests have been done with diluted IPA at atmospheric pressure, the contact times were recalculated considering the respective reaction temperatures. The contact time is calculated assuming the density of the catalyst as unity when it is not provided, and assuming that all gas flow rates (isopropanol and inert gas) are given in NTP conditions when it is not specified. Finally, it was calculated assuming only pure catalyst and no diluent (inert solid) in the catalyst bed, as these conditions are often not provided. The contact times are all in the range of 0.02 to 0.7 s.

**Table 2.** Selection of recent isopropanol dehydration catalysts and reaction protocols resulting in higher conversion rates and propylene selectivity.

| Catalysts | [b] Active Sites Possibly Responsible for Propylene | Reaction Conditions | IPA Conversion (%) | Propylene Selectivity (%) | * Other Products | Ref. |
|---|---|---|---|---|---|---|
| $SiO_2$-$ZrO_2$ | LAS BAS | T = 180 °C $P_{IPA}$ = 3 KPa $N_2$ flow rate = 48.5 mL·min$^{-1}$ [a] $t_{cont}$ = 0.07 s | 92 | 100 | DIPE [catalysts with Si/(Si + Zr) ratios other than 0.7] | [29] |
| Zr-SBA-15 | N.A. | T ~275 °C IPA feed rate = 11 g·h$^{-1}$ He flow rate = 833 mL·min$^{-1}$ [a] $t_{cont}$ = 0.03 s | 96 | 100 | Acetone [T > 275 °C] DIPE [T ~200 °C] | [30] |
| Pd/AlGa-x | LAS | T = 250 °C $P_{IPA}$ ~ 3.9 KPa $N_2$ flow rate ~65 mL·min$^{-1}$ [a] $t_{cont}$ = 0.02–0.05 s | > 95 | 97 | DIPE [T ~ 150 °C–178 °C] Acetone [catalysts with Ga/Al = 0] | [31] |
| Galium borates | BAS | T = 300 °C IPA & $N_2$ flow rate = N.A. [a] $t_{cont}$ = 0.13 s | 100 | 98 | Acetone [T > 300 °C] | [32] |
| $SiO_2$-$ZrO_2$ | BAS | T > 210 °C $P_{IPA}$ = 3 KPa $N_2$ flow rate = 48.5 mL·min$^{-1}$ [a] $t_{cont}$ = 0.07 s | ~90 | 100 | Acetone [over catalysts with Si/(Si + Zr) ratios < 20%] | [33] |
| Zr-KIT-6 | LAS | T = 300 °C IPA & $N_2$ flow rate = N.A. [a] $t_{cont}$ = 0.5 s | 93 | >98 | DIPE | [34] |
| Modified zirconium phosphates | BAS LAS | T > 200 °C $P_{IPA}$ ~4.3 KPa He flow rate = 20 mL·min$^{-1}$ [a] $t_{cont}$ = 0.53 s | 100 | 100 | Acetone DIPE [Mo & In doped catalysts] | [35] |
| Pt-Pd-supported $Al_2O_3$ | weak acid sites | T = 300 °C IPA feed rate = N.A. [a] $t_{cont}$ = N.A. | 100 | 100 | DIPE [mainly formed on Pt/$Al_2O_3$ catalysts] | [36] |
| Ni-W sulfides | N.A. | T = 250 °C $P_{IPA}$ ~ 5.9 KPa Diluent gas = 41.7 mL·min$^{-1}$ [a] $t_{cont}$ = 0.07 s | 100 | 100 | — | [37] |
| Ag NPs/mesoporous silica | BAS | T = 250 °C $P_{IPA}$ = 1.2 KPa $N_2$ flow rate = 60 mL·min$^{-1}$ [a] $t_{cont}$ = 0.05 s | 100 | >95 | Acetone [in $O_2$ atmosphere] | [38] |
| Zeolites | N.A. | T = 175 °C IPA feed rate = 15 g.h.mol$^{-1}$ $H_2$ flow rate = 30 mL·min$^{-1}$ [a] $t_{cont}$ = 0.67 s | 99 (Hβ) 90 (HY) | 100 (Hβ) 99 (HY) | DIPE [contact time < 0.2 s] | [39] |

* Different catalyst composition/structure, reaction temperature, contact time than the optimized conditions for propylene production. Exact conditions resulting in DIPE and acetone are given in square brackets []; [a] contact time ($t_{cont}$) calculated at the reaction conditions assuming the catalyst density as 1.0 g·mL$^{-1}$; [b] active sites that are beneficial for high propylene yield/selectivity using the reported catalysts at different conditions employed. These sites could be specific for each reported catalyst and may vary with catalysts/experimental conditions. N.A. = sufficient information is not available; **_N.B._** = all the tests performed in fixed-bed reactors at atmospheric pressure; LAS = Lewis acid sites; BAS = Brønsted acid sites; IPA = isopropanol; DIPE = diisopropylether.

This implies that the reactor volume could be about 30 times smaller, which can have a huge impact on the investment cost. In that case, the main challenge could be also to provide enough power (heat) to the reaction to carry it in industrial conditions. The dehydration reaction is endothermic, so heat has to be provided and the faster the reaction, the higher the power to be applied on the reactor.

For example, the amount of heat required from 200 °C is around 51 kJ·mol$^{-1}$, so a 60 kt IPA conversion unit may require about 2 MW. If the contact time is very short, the catalyst bed is also short, and high heat transfer reactors such as multitubular fixed-bed reactors or fluid bed reactors would have to be used.

## 4. Isopropanol Dehydration for Industrial Processes

### 4.1. Mitsui Patented Process

#### 4.1.1. General Process Conditions

Mitsui is said to have implemented the IPA dehydration in its phenol process, but surprisingly, it was not reported in their press releases. Only recently Mitsui started to communicate on IPA dehydration to propylene, but for production of biobased propylene.

The advantage in the phenol process is to avoid generating acetone as a coproduct, when the two products have disconnected markets [40].

An advantage of the recycling process is that the propylene, which is generated by dehydration, does not carry the usual contaminants of the petrochemical propylene. For example, the sulfur compounds such as COS (carbonyl sulfide) or arsenic as heavy metal contaminant, which inhibit the cumene synthesis catalyst have been removed in the loop (hydrogenation/dehydration). Therefore, less deactivation can be expected when using recycled propylene than when using petrochemical propylene. This advantage of lower contaminants would also be relevant for cases other than phenol synthesis.

In the process, an acidic catalyst is generally used. Although the patents claim many catalysts, the preferred ones are γ-alumina and titania. The dehydration reaction is preferably done between 260 and 350 °C, and under a pressure such as 1.8 MPa (or 18 Bars). The reaction products are then cooled below 40 °C. In those conditions, liquid propylene can be recovered and further dried and purified before use in the cumene synthesis.

The dehydration process could use a fluid bed reactor, but a fixed-bed unit is preferred. A fluid bed catalyst should be in powder form, with no diffusion limitations, while a fixed-bed catalyst should have a particular shape to reduce the pressure drop. A fluid bed reactor allows a continuous catalyst regeneration or replacement, while a fixed-bed reactor has a simpler design. So, the catalyst lifetime (or cycle time) is an important criterion in the selection of the fixed-bed operation. Propylene purities above 99.5% are claimed to be achieved.

In a practical example, a tubular reactor of 25.4 mm internal diameter and 500 mm length was loaded with 20 mL of alumina catalyst with particle size between 1.19 and 2.38 mm. When the reactor reaches 320 °C, IPA is fed at a 40 mL·h$^{-1}$ rate at a pressure of 1 MPa (10 Bars). The contact time can be calculated as: $1/(40 \times \text{density}(0.786) \times /\text{MM}(60) \times 22{,}400 \times (320 + 273)/273 \times (1/10)/20/3600) = 28$ s). The reaction products are liquid water (mostly) and gaseous propylene, respectively 9.5 g·h$^{-1}$ and 12.2 L·h$^{-1}$. The IPA conversion rate was 99.6% and propylene yield was 99.3%. The propylene purity in the gas phase was 99.9%.

### 4.1.2. Preferred Catalysts: Alumina

In a separate patent family, Mitsui patented a proprietary alumina [41]. The key features of that γ-alumina is a low content of alkaline metals (less than 0.5 wt.%), and less than 10 wt.% silica. The catalyst has a low acidity, quantified by Hammett indicators, in the pKa range between 3.3 and 6.8, with less than 0.5 meq·g$^{-1}$ (dry basis). In addition, the alumina should have a mean pore diameter between 3 and 15 nm, and preferably 4 to 7. Correspondingly, the pore volume is preferably between 0.5 and 0.8 mL·g$^{-1}$.

### 4.1.3. Acidity Scales

Solid acid catalysts are often characterized with the Hammett acidity scale. This is very convenient in patent applications because it is based on color indicators, and in case of dispute with a competitor it would be easy to convince a judge that the color with a given catalyst is not the same as with a competitor's catalyst. Besides this advantage, there are several drawbacks: the acidity is usually measured with the indicator in benzene solution (other solvents are also possible), but the amount of water on the solid is difficult to control, and the adsorption of solvents and titrants on the catalyst surface are not sufficiently controlled. It is sometimes possible to directly adsorb the color indicator on the catalyst surface from gas phase, but this is limited to white catalysts.

Acidity is also measured by ammonia thermo-programmed desorption (TPD), but this gives only a rough indication of the strength and number of acid sites. It can also be measured through calorimetric measurements in which the energy of adsorption for each dose of ammonia is related to the strength of the acid sites. This is much more precise but also much more time consuming.

Acidic resins cannot be characterized without water, and their thermal stability limits the potential use of other techniques. Looking at the various acidity scales and catalysts that have been characterized, a Hammett acidity between 3.3 and 6.8 corresponds to a low-acidity catalyst. Other products in this range also include silicalite and some alkaline forms of zeolites. Acidic resins such as Amberlyst 15 and Amberlyst 35 are more acidic. Silica-alumina and acidic zeolites are also much more acidic than aluminas. So, the good performance of alumina in this reaction indicates that one does not need a strongly acidic catalyst. However, low basicity would be more relevant to avoid the side reactions leading to acetone and DIPE.

### 4.1.4. External Mass Transfer Limitations

The challenge with propylene is that it is more reactive than ethylene and isobutene and that it could polymerize and deactivate the catalyst leading to a loss in yield. The temperature and residence time, at which the reaction is carried out, are equally important. They determined a maximum residence time based on the linear velocity for the IPA, which has to be above 1 cm·s$^{-1}$ and preferably below 10 cm·s$^{-1}$ in the reaction conditions (pressure and temperature). An excessively high linear velocity would generate more pressure drop, and certainly an excessively short contact time in a given reactor. When the linear velocity is too low (below 1 cm·s$^{-1}$), the conversion drops. This is an indication of external mass transfer limitations taking place at low gas velocities. In those conditions, when the temperature is increased further, conversion also increases but byproducts are significant at low temperature.

In Table 3, extracted from Mitsui's patent, the experimental data on shaped catalysts are reported. When looking at the values for "comparative examples" three and five (CE3 and CE5), the volume of catalyst and the IPA flow rate have been multiplied by five. The contact time remained the same in both cases, but the gas linear velocity in CE5 is five times that of CE3. Similar conversions and yields are reached at a lower temperature. There is a further increase in conversions/yields when the linear gas velocity is further increased in examples E12 and E13. In addition, although the amount of data is very limited, looking at the first two data of CE3 that are not at full conversion, the calculated activation energy is about 27.5 kJ·mol$^{-1}$, which is in line with the usual values resulting from external mass transfer limitations.

**Table 3.** Data extracted from Mitsui's patent EP379803 illustrating the external mass transfer limitations.

| Reaction Conditions | Comparative Example | | | | | | | | Example | | | |
| | CE3 | | | CE4 | | CE5 | | | E12 | | E13 | |
|---|---|---|---|---|---|---|---|---|---|---|---|---|
| Catalyst loading (mL) | 20 | | | 20 | | 100 | | | 200 | | 400 | |
| Reaction pressure (kg/cm$^2$ G) | 20 | | | 10 | | 20 | | | 18 | | 18 | |
| Isopropanol feed (mL/h) | 60 | | | 60 | | 300 | | | 600 | | 1200 | |
| $(M \times R \times T)/(3.6 \times P \times \pi \times r^2)$ | 0.1 | | | 0.2 | | 0.5 | | | 1.1 | | 2.1 | |
| Catalyst center temp. (°C) | 320 | 340 | 360 | 300 | 320 | 280 | 290 | 310 | 280 | 290 | 280 | 290 |
| **Results** | | | | | | | | | | | | |
| Isopropanol conversion (mol%) | 80 | 96 | 99 | 81 | 99 | 59 | 88 | >99 | 73 | >99 | 71 | >99 |
| Propylene selectivity (mol%) | 87 | 96 | 99 | 70 | 95 | 84 | 95 | >99 | 90 | >99 | 88 | >99 |
| Propylene yield (mol%) | 70 | 92 | 98 | 57 | 94 | 50 | 84 | >99 | 66 | >99 | 62 | >99 |
| By-product * amount (mol%) | 13 | 4 | 1 | 30 | 5 | 16 | 5 | <1 | 10 | <1 | 12 | <1 |
| Propylene amount (mol/h) | 0.55 | 0.72 | 0.77 | 0.45 | 0.74 | 1.95 | 3.28 | 3.85 | 5.16 | 7.69 | 9.81 | 15.39 |

* By-products include acetone, diisopropyl ether, etc. For calculation, isopropanol (molecular weight 60) has a density of 0.785 g/cm$^3$.

The experimental results were generated with catalyst particles between 1.19 and 2.38 mm diameter. Typical experimental conditions were 1.0 MPa (relative pressure), 320 °C, and a liquid hourly space velocity of 3 h$^{-1}$. The reactor was the same as in the previous patent application. The IPA conversion reached 98.4% and the propylene yield 98%. The propylene purity is 99.5% and the main side product is acetone (0.5%). In other examples, the byproducts also include DIPE.

### 4.1.5. Internal Mass Transfer Limitations or Capillary Condensation

The results also show that for the catalysts with low pore diameters, the IPA conversion under pressure starts to decrease, which suggests that capillary condensation might have occurred, and/or that there are internal mass transfer limitations.

The relationship between catalyst activity and the pore diameter is determined using the alumina with very low sodium content in the following experimental conditions: 1.8 MPa (relative pressure), 300 °C, a liquid hourly space velocity of 3 h$^{-1}$ and a contact time of about 35 s. The results reported in Figure 4 are Table A and Fig. 1 taken from the patent. When the pore diameter is too large, conversion is low because of the low surface area. When the pore diameter is below 6 nm, the conversion might be limited by mass transfer.

Table A

| No. | Mean pore diameter(Å) | Conversion of isopropanol (mol %) |
| --- | --- | --- |
| 1 | 31 | 86.8 |
| 2 | 40 | 87.1 |
| 3 | 62 | 86.8 |
| 4 | 75 | 74.9 |
| 5 | 114 | 68.9 |
| 6 | 150 | 59.1 |
| 7 | 160 | 52.2 |
| 8 | 190 | 27.1 |

Fig. 1

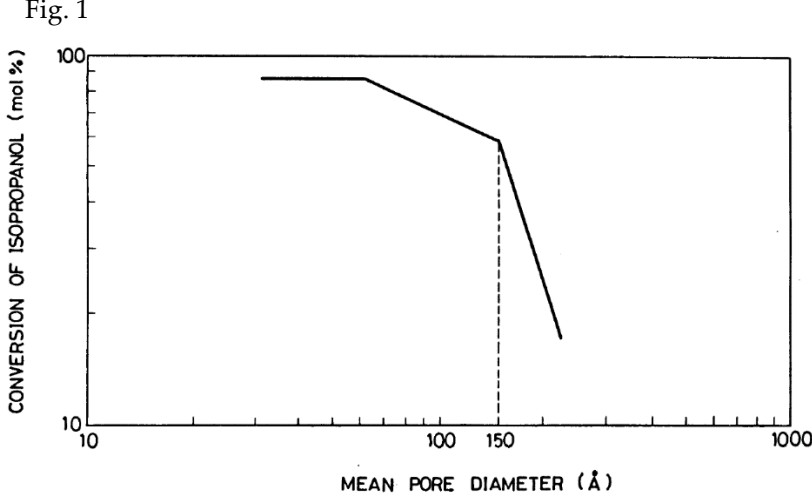

**Figure 4.** Table A and Fig. 1, extracted from patent EP379803. Impact of catalyst pore diameters.

When the pressure decreases, the IPA conversion increases and the propylene yield increases. So, one could be tempted to operate in these conditions, but the separation of the products becomes more complicated. In order to keep a low operating temperature in the range of 300 °C, it is necessary to have a high gas linear velocity. This means that the reactor length should be sufficiently long to get enough residence time.

Using the Laplace–Kelvin equation, we can calculate the pore diameter at which capillary condensation is expected to occur. Water and IPA are the potential candidates for this phenomenon, but as the reaction proceeds IPA is more likely to condense near the reactor entrance, while water partial pressure is larger near the end of the reactor. Taking into account the surface tension and respective partial pressures, IPA is the reagent on which the calculations should be done. In Figure 5, the data are reported for 18 bars,

1 bar, and 0.1 bar. When the pore diameters become too small the equation is no longer valid, so it is reported here only for illustration purposes. At 18 bars and 200 °C, capillary condensation is to be expected for pores below 9 nm and below 4 nm when the reaction temperature is increased to 235 °C. As the total pressure decreases, capillary condensation is less likely. In Figure 5, we also reported typical pore diameters for some commonly used acid catalysts. Capillary condensation explains why solids such as alumina and titania proved to be more efficient than zeolites under pressure, and the acidity of these solids alone would not be the sole explanation.

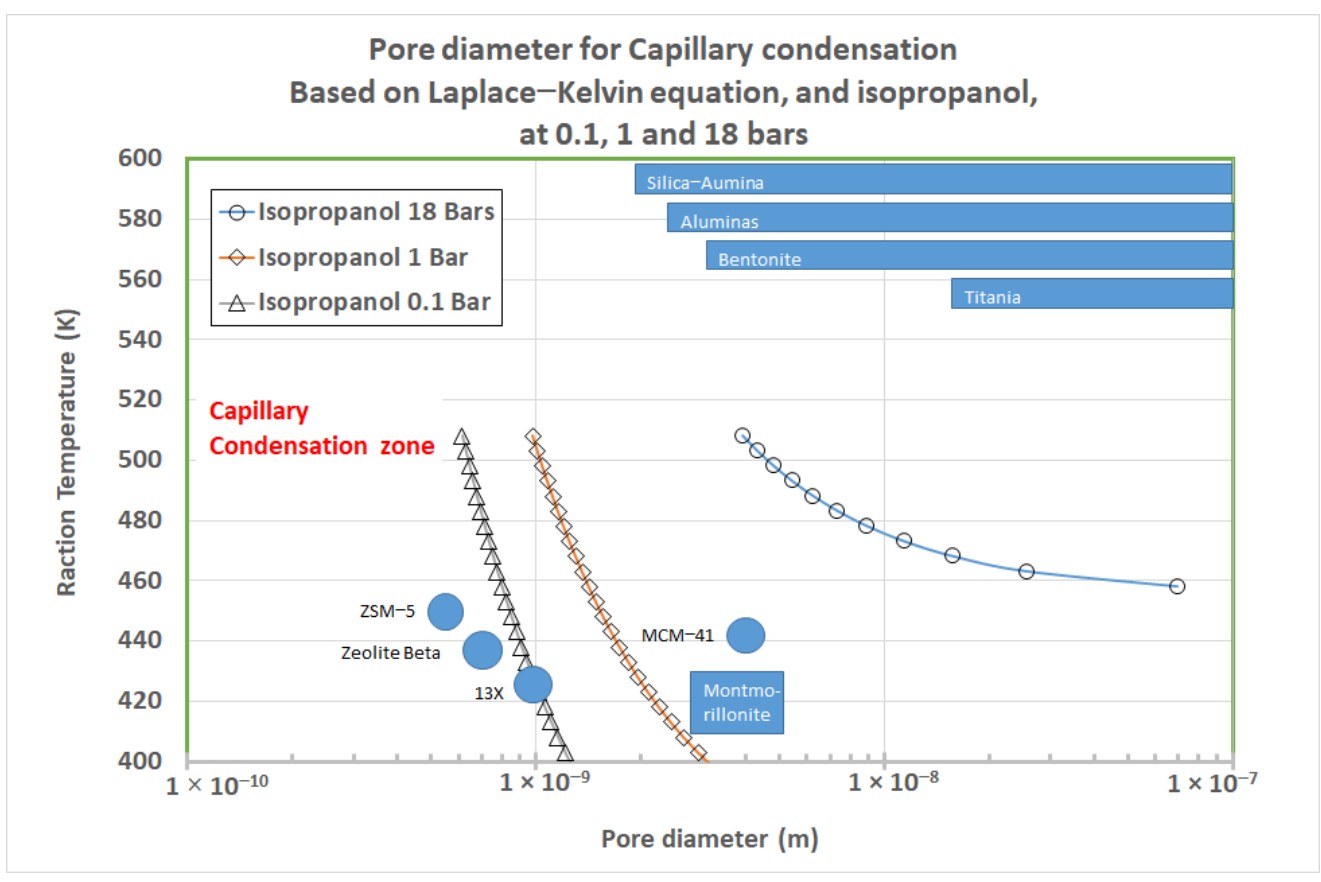

**Figure 5.** Calculation of the pore diameter for capillary condensation based on the Laplace–Kelvin equation for isopropanol. Pore diameter ranges for several candidate catalysts are also reported.

There is a compromise between the surface area and the pore diameter, because large pores lead to low surface area. If only diffusion limitations are feared, it is possible to overcome this issue by using catalysts with bimodal pore size distribution. Usually, internal mass transfer limitations are not evidenced in academic experiments involving crushed catalysts, and the impact of the pore size distribution required to use shaped catalysts of a few mm size.

4.1.6. Catalyst Deactivation

Catalyst deactivation studies showed that although IPA conversion remained above 90% after the catalyst is loaded with 13 wt.% of coke, coke deactivation can be expected for these catalysts and in these reaction conditions.

In a subsequent patent application, Mitsui describes further improvements to the catalysts [42]. The alumina should have low impurity content (less than 0.3 wt.% excluding silica). More specifically, it should have a low sulfur (as sulfates) content and a low sodium content. The reported catalyst contains between 0.5 and 5 wt.% silica. The catalyst acid–base properties are certainly affected, not only by the presence of alkaline impurities and

sulfates but also by the silica. As discussed previously, basic sites catalyze the formation of side products, while overly acidic sites can trigger coke formation.

Since the reaction is going to generate a lot of water/steam, the catalysts have to be resistant to steaming conditions. The presence of these impurities will severely affect the resistance to steaming, and gamma alumina was partially converted into alpha alumina. The activity of the catalyst is reduced and the selectivity to side products increased. This suggests that DIPE is an intermediate that is, for example, produced by the reaction of IPA with propylene, which can further dehydrate to propylene.

### 4.2. IFPEN Patented Process

4.2.1. General Process Conditions

IFPEN also considered the dehydration of IPA in an ethanol stream [43]. Alumina is again the preferred catalyst with a low content of sodium and sulfur. The preferred conditions include an operation under pressure (0.2 to 2.0 MPa), temperature from 350 to 500 °C, and a weight hourly space velocity (WHSV) between 1 and 10 h$^{-1}$. Assuming the alumina packing density to be 1, the contact times calculated are in the range of 40 s (between 7 and 80 s) if pure IPA was used and comparatively less if water is co-fed with the ethanol–isopropanol–water mixture. Dehydration of an alcohol mixture is particularly relevant when it was produced either from syngas or through the IBE fermentation process.

4.2.2. Impact of Water Fed with Alcohol

The dehydration reaction is endothermic, but energy saving can be made to vaporize the reagents when recovering the latent heat contained in the reaction products. It may seem counterintuitive to make a dehydration reaction in the presence of water/steam; however, water can have several roles in the reaction: (i) it is a reaction product that favors a reverse reaction, (ii) it has a high calorific value and brings heat to the catalyst for an endothermic reaction, and (iii) it favors coke removal and contributes to keep the catalyst surface clean.

For the energy balance, we have to take into account that we have to bring energy to evaporate isopropanol. The amount of energy needed corresponds to the latent heat and the sensible heat. A significant part of it can be recovered after the reaction when the products have to be cooled to room temperature. If water is added with isopropanol, the energy needed to evaporate it will be mostly recovered after the reaction, provided that an efficient combination of heat exchangers is implemented.

Isopropanol's enthalpy of vaporization is around 40 kJ·mol$^{-1}$, near its boiling point [44], while that of water is around 41 kJ·mol$^{-1}$ at the same temperature. So, there is enough energy in the product stream (due to steam) to evaporate the feedstock (isopropanol). It also means that adding water to the reagents, as described in this patent application, has a minor impact on the energy consumption. Water/steam is used as a heat carrier in endothermic dehydration reactions, which allows the use of adiabatic reactors. However, the water in the feed stream reduces the IPA's partial pressure and favors the reverse reaction, but it can also inhibit more side reactions, leading to a better global selectivity.

4.2.3. Catalysts and Impurities Generated by the Reaction

A major difference with the Mitsui process concerning the catalyst is that the catalyst phase is not restricted to gamma alumina, but other phases can also be present in the catalyst. The typical reaction conditions are 375 °C, 0.2 MPa, a mix of ethanol–IPA–water (1/1/2 weight ratios) and a calculated contact time of 1.8 s. Depending on the process conditions and catalysts, the following impurities are detected: DIPE, propane, propanal, and ethoxypropane (because of co-fed ethanol and IPA). DIPE is the intermolecular dehydration product and propanal is the n-propanol dehydrogenation product. If propanol was not present as impurity in the feed stream, it is probably produced by rehydration of propylene to 1-propanol followed by dehydrogenation to propanaldehyde. The hydrogen, which is either produced by this dehydrogenation reaction or generated by coke formation,

can be also transferred to propylene to generate propane. Propanal and acetone can lead to aldol reaction products that may remain trapped in the catalyst and slowly deactivate it. So, it is relevant to track those impurities even at low dosage.

### 4.3. Versalis' Patented Process

#### 4.3.1. General Process Conditions

Versalis also considered IPA dehydration to integrate it in a phenol process [45]. In their case, the preferred catalysts are large pores zeolites, having BEA or MTW structures such as ZSM-12.

The operating conditions differ from the previous patents, as the preferred temperature range is lower, i.e., between 175 and 220 °C, and the pressure range is between 0.01 and 2 MPa, preferably between 0.09 and 0.2 MPa, which are lower when compared to Mitsui's process. A pressure slightly above the atmospheric pressure is necessary to push all the reagents and products through the unit and downstream equipment. The WHSV is in the same range as that of the IFPEN case for similar pressures. The proposed catalysts are shaped, for example, into $2 \times 10$ mm extrudates with an alumina binder. The examples were generated for temperatures close to 200 °C, pressures between 0.1 and 0.2 MPa, and WHSV between 1 and 2 $h^{-1}$. This translates into a contact time of about 3 to 11 s. The low total pressure implies that the reaction products must be cooled to low temperature (5 °C) to trap most of the water.

#### 4.3.2. Deactivation and Side Products

The side products reported in this reaction include DIPE but also isobutane, acetone, olefins, and four to six carbon paraffins. With a ZSM-12 catalyst, a conversion above 99.0% and a selectivity above 97.9% were recorded over 1008 h. The selectivity to DIPE and other side products tend to decrease over the time of operation. This might be a sign that non-selective sites tend to be deactivated faster than the selective sites. With a Beta zeolite, similar conversions and yields are obtained. Alumina, at 0.1 MPa, for which a higher temperature is required, leads to slightly higher conversion and propylene selectivity. In similar conditions, alumina requires about 50 °C more operating temperature to achieve similar performance to MTW and BEA zeolites. However, these zeolitic catalysts have not been tested in the high-pressure range that would facilitate the downstream separation.

Formation of saturated hydrocarbons (propane), either from hydrogen generated from side reactions or from hydrogen transfer during coke formation, is not necessarily an issue for all propylene applications. Only the "Polymer Grade" propylene has a low propane content, whereas the "Chemical Grade" propylene can accept several percent of propane. However, the formation of this side product is accompanied by a faster deactivation of the catalyst, which might require frequent catalyst regeneration/decoking and induce the selection of specific reactor configurations (such as circulating fluid bed reactors or tandem reactors).

### 4.4. Reverse Reaction: Propylene Hydration

Propylene hydration is currently the alternative process for acetone hydrogenation to produce IPA. As of 2022, it represents about 2/3rd of the installed capacities for IPA production. The reverse reaction, i.e., propylene hydration to IPA, deserves attention, as the IPA dehydration is equilibrium limited.

#### Mobil Oil Process

Mobil Oil Corporation patented a process [46] in which propylene reacts with an under-stoichiometric amount of water over metallosilicate catalysts to produce IPA and DIPE. Preferred catalysts include ZSM-5, ZSM-23, ZSM-35, and Ferrierite. Water concentration is limited to avoid deactivation of the catalyst. So, the water to propylene ratio is between 0.05 and 0.499, preferably between 0.18 and 0.33. However, if water deactivates or reduces the activity of the catalyst during the hydration reaction at high IPA concentration, it will

be also the case during the dehydration reaction, especially at high conversion rates where a lot of water is produced.

Equilibrium considerations suggest that the hydration reaction is favored at low temperatures, high pressures, and high water/propylene ratios. At low water concentration, DIPE would dominate the equilibrium composition. The challenge is that at high water concentration the catalyst is unstable, and at low water concentration deactivation by coke formation would occur. Hence, it is necessary to operate at a low conversion per pass, preferably at temperatures in the range of 150 to 250 °C, pressures between 2.8 and 7.0 MPa, and WHSV's between 1 and 10. These conditions are similar to those described in the dehydration processes, which suggests that the reverse reactions are taking place. However, the examples are best in the 150 to 200 °C range but with low productivities, and in several cases both IPA and DIPE are produced.

## 5. Intentional Diisopropylether Synthesis

### 5.1. Patented Processes

Several patents and publications report the intentional synthesis of the byproduct DIPE, which should be avoided when propylene is the targeted product. They give valuable information about the mechanism of DIPE formation, and the conditions that should be avoided when propylene is the target. DIPE could be used as a fuel additive in gasoline, and that is probably the reason why most of the patents were filed by oil companies. A significant share of DIPE was produced as a coproduct in propylene hydration to IPA.

### 5.1.1. Gulf Oil Process Conditions

As explained in the Gulf Oil patent [47], the hydration is catalyzed by acid catalysts, and both reactions from propylene to IPA and DIPE are equilibrium limited. In addition, the dehydration of IPA to DIPE can also take place over acid catalysts which are also limited by the equilibrium. The alcoholysis of propylene with IPA is also possible, and does not imply water formation or consumption, but it is equilibrium limited and catalyzed by acids.

In the process described in this patent, DIPE is produced from IPA, and eventually some propylene but in a liquid phase and under pressure. The preferred solid acid catalyst is a sulfuric acid-treated Montmorillonite with sufficient acidity to maintain a pH below 4 in water at standardized conditions. The reaction is done in a fixed-bed reactor at temperatures between 120 and 250 °C, and more preferably between 185 and 210 °C. In the first example, the conditions employed are 110 atm and a liquid hourly space velocity (LHSV) of 10 h$^{-1}$ (based on the liquid flow rate at room temperature) that corresponds to about 6 min of equivalent contact time. In the best conditions, IPA conversion and DIPE selectivity are in the range of 50–60%.

Alternative catalysts such as acidic resins such as the Dowex 50WX8, Amberlyst A15, and AGC-243 were also tested, but at lower temperatures because of their low thermal stability, and their productivities were lower. Acidified 13X molecular sieve and Silica-Alumina 979 from Grace were found inadequate but without details. However, when propylene is co-fed on the 13X catalyst, a propylene yield above 100% is achieved, suggesting that IPA dehydration to propylene is taking place instead. Other inorganic acids also gave positive propylene yields. However, they might be attractive for IPA dehydration to propylene.

When propylene is co-fed with IPA, a part of it is also converted. This illustrates that the reactions are indeed limited by equilibrium and/or that propylene hydration or alcoholysis are taking place.

These conditions are relevant for IPA dehydration to propylene as described in the previous section when the catalyst and process favor capillary condensation. Then, in the small pores, similar conditions of pressure and temperature could be found, which can explain some DIPE formation. Based on the patent, the most appropriate conditions for DIPE production is a temperature around 200 °C and a pressure between 35 and 110 atm while operating at low contact times. The pressure is above the range previously described

for IPA dehydration to propylene and the temperature is lower, while the contact time is longer.

### 5.1.2. Chevron Process Conditions

Chevron [48] patented a process for the hydration of propylene in which the DIPE is also hydrolyzed with excess water. The key idea in this process is that the DIPE is recycled at the front-end of the hydration process, where it is contacted with the process water in large excess. The reaction produces IPA, and then the stream is merged with the fresh propylene. The products are separated and remaining DIPE is returned at the front end. If we assume that the reaction reaches a steady state at this second stage, it means that there is no accumulation of DIPE in the process. DIPE is more volatile than water and IPA, so it can be separated from them as an azeotrope containing 4% IPA and 5% water. However, IPA separation from water is more difficult because of the azeotrope. The interest for the refiners is of course to use "Refinery Grade propylene", which contains a lot of propane but acts as inert in the reaction.

In the Chevron process, the preferred catalyst is an acidic resin for both the reaction stages. The acidic resins were Amberlyst XN1011 or Amberlite XE-372. However, other solid acid catalysts such as ZSM5, silicalite, alumina-silica, or alumina were also used. The pressure is about 100 bars and the temperature between 150 and 260 °C, while the reaction is preferably done with an excess of water. The impurities in the collected IPA come partly from the olefinic stream that contains some C4s and also from the dimerization of propylene. The LHSV in the first reaction zone is between 0.03 and 1.0 h$^{-1}$, indicating a contact time of several hours. In the examples, the DIPE reacts with water in a 1/24 ratio over an acidic resin at about 150 °C and 100 bars. Equilibrium was achieved at a low LHSV (more than 20 h contact time), and gave a composition in equivalent propylene of 6% propylene, 22% DIPE, and 72% IPA. So, the equilibrium constant estimated in these conditions is 0.017, which suggests that a high amount of water is needed to shift the equilibrium or that completely different temperature and pressure conditions should be selected.

### 5.1.3. Mobil Multistage Process in Absence of Aqueous Phase

More recently, Mobil patented a process for the production of IPA and DIPE through propylene hydration with a Beta zeolite catalyst [49]. Propylene is fed along with water, IPA, and/or DIPE in a multiple stage reactor, and with increasing "water equivalent" (water, IPA, DIPE) ratios.

Figure 6 illustrates the composition reached at each stage with intermediate addition of propylene, water, and IPA/DIPE at about 70 bars and 160 °C. If the same equilibrium was reached at each stage, then the equilibrium constant that can be calculated from these compositions should have remained constant, but that is not the case. The Q value is the ratio of water equivalent (water + IPA + DIPE) to the propylene equivalent (propylene + IPA + 2 × DIPE) at each stage.

However, the process favors the alcoholysis (etherification) of propylene with IPA over the hydration and maintains a non-aqueous phase in the reactor. The organic phase remains rich in IPA, DIPE, and propylene. So, the process differs from the previous one in the sense that a small amount of water is present and either DIPE or IPA is recycled in the reactor depending on the targeted product.

The preferred reaction conditions are 120 to 220 °C, 34 to 138 bars, and a water to olefin ratio of 0.1 to 5. In addition, the main feature of this process is the absence of an aqueous phase, which is favorable to reduce the ageing of the catalyst. Catalyst ageing is favored by water and propylene phases. For that reason, water is replaced by IPA in the hydration of propylene to DIPE. Several catalysts were screened in autoclave and zeolite beta showed the best intrinsic selectivity at low conversion compared to ZSM5 > Amberlyst 15.

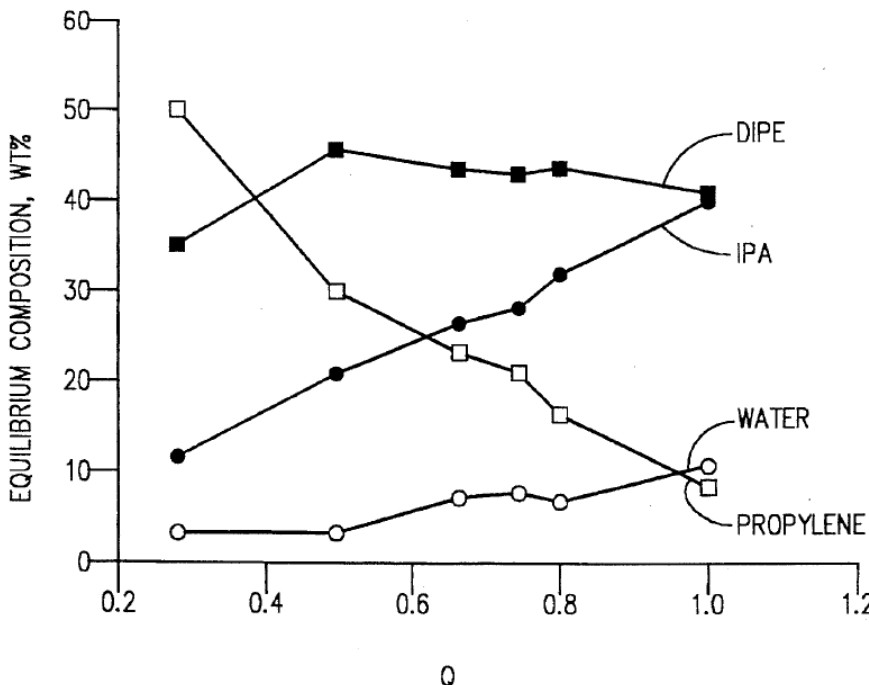

**Figure 6.** Extracted from Mobil patent for a multistage process with increasing water equivalent.

When DIPE is fed with an equal molar amount of water to a fixed-bed reactor, the conversion favors the formation of IPA over propylene, at a low conversion. Since the experiments are done at high WHSV, meaning short contact time, they illustrate the initial kinetic of the reactions. In presence of water, DIPE hydrolysis is favored over DIPE decomposition to propylene.

Other experiments show that at 160 °C and about 70 bars, DIPE is converted to IPA at a low contact time. The equilibrium constant at 162 °C, calculated from the composition at equilibrium, was found to be 2.68. The value is consistent with the favored formation of DIPE at low temperature from propylene and water and under pressure, but is valid only for single liquid phase systems.

In the patent application already discussed above [46], Mobil gives more details on the preferred catalyst, zeolite Beta. The production of DIPE is carried from propylene and water or from IPA, and the surface acidity of the catalyst is reduced by a chemical treatment with a diacid such as oxalic acid. However, dealumination of the zeolite should not go beyond 50%. When the porosity of the zeolite is filled with an organic compound, the dealumination process affects mostly the external surface of the zeolite crystals and reduces the acidity. With such a treatment, the activity for IPA conversion is increased. There is an increase in the catalyst activity, but no drastic change in product distribution is observed at similar conditions (about 160 °C and 70 bars).

*5.2. Academic Literature*

In the academic literature, the publication from Heese et al. [24] provides additional inputs on the conditions leading to DIPE or propylene from IPA. The authors conclude that in both the gas and liquid phases, the reaction proceeds by two parallel routes: the dehydration of IPA and the alcoholysis (etherification) of propylene, with activation energies of 75 and 92 kJ·mol$^{-1}$, respectively. The formation of DIPE is favored at high pressure and sufficient temperature to speed up the reaction. For the liquid phase experiments, the reaction was carried at 50 bars and 120 °C with an acidic resin in trickle flow mode. In those conditions, propylene is in supercritical conditions, besides a liquid water phase. IPA is the primary product, but at longer contact times DIPE production dominates, as propylene and IPA are consumed.

When pure IPA is fed in the same conditions (50 bars, 120 °C), propylene, water, and DIPE are the primary products, and at longer contact times, IPA reacts with propylene to form more DIPE. At atmospheric pressure and the same temperature, the same products are formed; however, this time, propylene and water dominate the products, and although DIPE is formed at low contact time, it disappears completely. After about 13 s contact time (based on GHSV), the conditions are close to equilibrium, which seems to be fully reached by 2 min contact time.

The three equilibrium reactions: propylene hydration to IPA, IPA dehydration to DIPE, and propylene alcoholysis to DIPE take place in the gas phase as well as in the liquid phase.

## 6. Catalyst Selection Criteria

Based on this literature review, some guiding principles for catalyst selection can be derived:

- Acidic catalysts are needed, with low to medium acid strength and with extremely weak basic sites to avoid side product formation;
- Pores of sufficiently large diameter are needed, especially if the reaction is carried out under pressure;
- Catalysts should be resistant to steaming conditions, since the reaction is producing water at a high temperature;
- A catalyst life of at least several months is needed for fixed-bed operations, and deactivation by coke formation or surface reconstruction should be minimized.

### 6.1. Capillary Condensation

Capillary condensation in the porous catalyst should be considered, especially if the reaction has to be carried out under pressure. In order to maximize the catalyst activity, a large surface area is preferred. This would come with small pore diameter in which reagents and products could accumulate and lead to deactivation. So, when working under pressure, catalysts with larger pore diameters might be preferred. When capillary condensation occurs, the pores and the surface are saturated with IPA and the reaction can take place in a supported liquid-like phase or in conditions that favor DIPE formation. Especially, it is more likely to have two neighboring adsorbed IPA molecules on the catalyst surface, thereby increasing the chances of producing DIPE. To reduce side product formation, larger pore diameters, higher temperature, and/or lower pressure would be favorable.

### 6.2. Catalyst Resistance to Steaming

Catalyst steaming will occur during the reaction because of the high water partial pressure and temperature, and this adds a selection criteria to the catalyst for industrial operation. Deactivation due to steaming will be more severe under pressure and at high temperatures, so the choice of operating conditions will have a large impact. Of course, the presence of high partial pressure of steam on the catalyst will impact the type of acid sites present on a working catalyst. All Lewis acid sites are likely to be quickly converted to Brønsted acid sites.

### 6.3. Catalyst Life Expectancy

If the catalyst deactivates quickly, it needs to be either reactivated or replaced. This has an impact on the choice of reactor technology and on the operating cost. If the catalyst deactivates in a matter of a few seconds but can be easily reactivated, the circulating fluid bed reactor is the technology of choice. The catalyst will be used as powder of a few tens of microns and internal and external mass transfer limitations can be disregarded.

If the catalyst deactivates in a matter of a few minutes or hours, then tandem fluid bed reactors can be the technology of choice. When the deactivation is a matter of hours or days, mobile-bed reactor technology is appropriate. The catalyst should be as shaped particles of a few millimeters, with high mechanical properties. In that case, the heat transfer during

the reaction would have to be done through the gas phase, meaning that, for example, the heat is carried by additional steam fed to the reactor.

Preferably the deactivation occurs in a matter of months and the reactor can be a fixed-bed reactor, and for example, a multitubular reactor to improve heat transfer to the catalyst. These reactors take time to be unloaded and reloaded with fresh catalyst, and any such period means a loss in production days and hence, a decrease in profit margin. So, the longer the catalyst life, the better it is. In a few cases, the catalyst can be reactivated in situ, but it might be preferable to unload the catalyst and make a better regeneration ex situ. This strongly depends on the catalyst cost, the deactivation mechanism, and the regeneration conditions.

### 6.4. Activation Energies and Reaction Selectivity

Basic sites on the catalyst surface catalyze the formation of side products such as acetone. In most of the application for propylene, acetone is a contaminant and has to be minimized.

A good catalyst should not only have a low formation rate of acetone, but also have a low activation energy, and more specifically a much lower activation energy than the IPA to propylene reaction. In fact, the larger the ratio of activation energies between propylene formation and acetone formation, the better it is. Any variation in reaction temperature will significantly affect the selectivity, and it would be easier to find more appropriate conditions for the reaction.

Figure 7 illustrates the impact of the reaction temperature on the selectivity for propylene production. For three different catalysts, the Arrhenius plots for the acetone and propylene production are reported, and they show that the activation energies for acetone production are lower than for propylene production. Alumina appears as an attractive catalyst since it has the lowest rate of formation of acetone. The data are extrapolated to 280 °C (1000/T = 1.8), where it is evident that niobium oxide would give similar selectivity than alumina. By increasing the temperature, the selectivity would be enhanced, and so a higher temperature would be preferred.

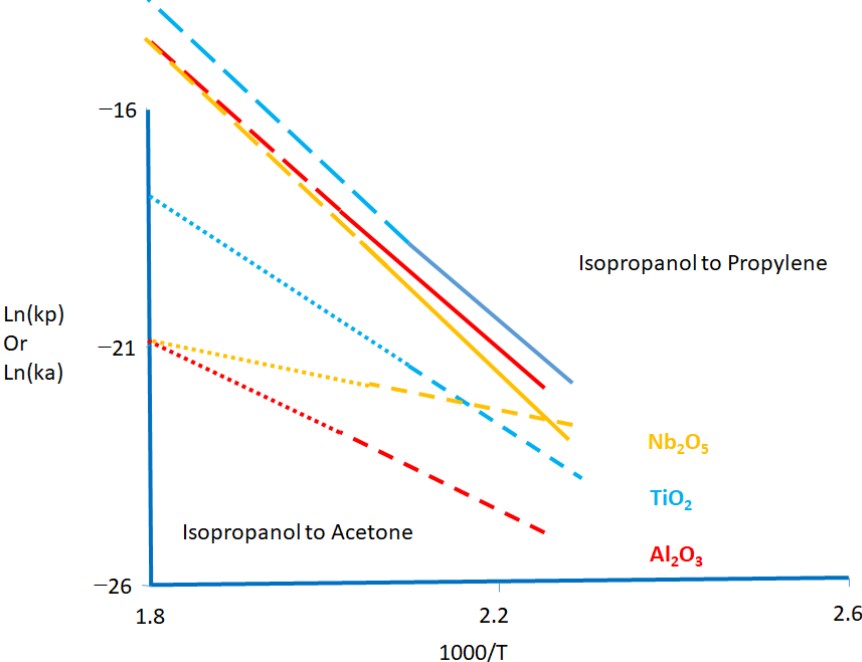

**Figure 7.** Arrhenius plots for acetone (dashed lines) and propylene (continuous lines) formation from isopropanol. Data computed from [50] and extrapolated to 280 °C.

Higher temperatures are likely going to lead to less acetone and less DIPE but are more favorable for catalyst deactivation by coke formation and/or surface reconstruction. Hence, an appropriate balance needs to be found for the selected catalysts.

## 7. Conclusions

This review highlights how the process conditions, and especially the downstream purification section, can impact the purity of recovered propylene and the required catalyst properties.

Isopropanol dehydration can be carried at atmospheric pressure or under pressure. Under pressure, the separation of products can be facilitated as water and the remaining IPA are condensed first upon cooling; while propylene will be condensed at lower temperature. This is particularly relevant if high-purity "polymer grade" propylene is needed in the process. Therefore, a total pressure of 20 bars and a reaction temperature of about 300 °C with contact times of several seconds would not be uncommon at an industrial scale. If traces of water, remaining IPA, acetone, DIPE, and propane can be accepted for the downstream process, then a reaction closer to atmospheric pressure might be suitable. That could be the case when "chemical grade" propylene is to be used. It is particularly relevant when the reaction products of the dehydration stage can be directly fed into the downstream reaction without purification, thereby saving significant operating and capital costs.

The choice of the operating pressure determines the type of catalyst. Higher pressure means higher temperature for the dehydration step. Some zeolitic catalysts would deactivate faster under the steaming conditions of the dehydration reaction. They are however appropriate for reactions around 200 °C, provided that they have large pores.

Several side products including DIPE, acetone, propanal, propane, olefins, and oligomers have been reported. They illustrate side reactions taking place: dehydrogenation, oligomerization, rehydration, hydrogen transfer, and probably aldolizations. Some of these impurities can be challenging in the downstream applications, and a "polymer grade" purity level might be difficult to reach.

In some cases, mass transfer limitations occur. In order to avoid this, a sufficiently high gas linear velocity is necessary. Additionally, catalysts with sufficiently large pores (pores larger than 4 nm, for example) are preferred when catalysts are used in the form of pellets. In addition, a bimodal pore size distribution can help to improve the diffusion in the catalyst particles.

Catalyst deactivation by coke formation, but also due to the formation of more stable phases in steaming conditions, has been reported. In addition, the selectivity for side products also changes with the time on stream. So, it is relevant to more deeply characterize the catalyst under working conditions.

Dehydration of IPA to propylene is favored in gas-phase conditions, while DIPE is more favored in liquid-phase conditions. The liquid phase is achieved under high pressure operation. Nevertheless, since capillary condensation can take place in the small catalysts pores, DIPE can be formed even in the gas-phase process if the catalyst is not properly selected.

The reaction might require high temperatures in order to obtain the appropriate adsorption–desorption equilibrium over the catalyst surface. At higher temperatures, the catalyst surface is not completely covered and adsorption of two IPA molecules or one IPA and one propylene molecule on two neighboring sites is less likely. So, the risks for the formation of DIPE are reduced. However, at higher temperatures, the formation of propylene oligomers becomes more favored.

**Author Contributions:** Conceptualization, J.-L.D. and A.A.; methodology, J.-L.D.; software, none.; validation, J.-L.D. and A.A.; formal analysis, J.-L.D.; investigation, J.-L.D., G.P., L.S.; resources, J.-L.D., G.P., L.S.; data curation, J.-L.D., A.A.; writing—original draft preparation, J.-L.D., G.P., L.S.; writing—review and editing, J.-L.D. and A.A.; visualization, J.-L.D.; supervision, J.-L.D. and A.A.; project administration, A.A, G.P. and J.-L.D.; funding acquisition, A.A. and J.-L.D. All authors have read and agreed to the published version of the manuscript.

**Funding:** The research presented here has been carried out under the EU project PyroCO₂. This project has received funding from the European Union's Horizon 2020 research and innovation programme under grant agreement No. 101037009 and results presented here reflect only the author's view and the Agency and the Commission are not responsible for any use that may be made of the information it contains.

**Data Availability Statement:** Not applicable.

**Conflicts of Interest:** The authors declare no conflict of interest.

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
