# Peer review of "Catalytic Dehydration of Isopropanol to Propylene"

_catalysts, doi:10.3390/catal12101097_

Round 1

Reviewer 1 Report

The paper is reviewing the catalytic dehydration of isopropanol to propylene, mainly from industrial point of view. It collects all the important technological parameters influencing the conversion and selectivity of the reaction circle. Summerizes the industrial processes, and gives hints for the selection of catlysts. It is logically structured and well written. The english style of the paper is good. It can be acceted as it is. 

Author Response

We thank the reviewer for his kind comments. We'll check again the english in the paper and spelling.

Reviewer 2 Report

Dubois et al. reported a catalytic dehydration of isopropanol to form propylene, which was used to characterize the acid-base properties of the catalyst. In some cases, high temperatures play a critical role in obtaining a proper adsorption-desorption equilibrium on the catalyst surface. As mentioned by the authors, at higher temperatures, the catalyst surface is not completely covered, which then reduces the risk of DIPE generation. As a result, the formation of propylene oligomers becomes more favorable. Overall, this approach is interesting and practical, while the means of characterization is comprehensive. The text provided is informative and well framed overall. Thus, I think this work is worthy of publication in Catalysts after the minor revisions:

1.      The abstract module is not very well written and does not highlight the role of catalytic dehydration of isopropanol to propylene in this paper.

2.      Some specific convincing experimental evidence needs to be provided, for example 4.2.2 should provide some data or graphs, which makes it easier for the reader to get some useful information.

3.      Note that these reactions essentially require experiments to be performed at high temperatures, which can make the experimental process potentially dangerous. Therefore, a more detailed experimental procedure needs to be provided for such high temperature experiments.

4.      The authors need to touch up the current version of the conclusion module as needed.

Author Response

We thank the reviewer for his kind comments.

1. The abstract module is not very well written and does not highlight the role of catalytic dehydration of isopropanol to propylene in this paper.

Several sentences have been added in the abstract to highlight the challenges.

2. Some specific convincing experimental evidence needs to be provided, for example 4.2.2 should provide some data or graphs, which makes it easier for the reader to get some useful information.

Paragraph 4.2.2 is about the effect of water content in the feed. It says that since water is in gas phase at the inlet of the catalytic bed and gas phase at the outlet there is no energy consumption for it. The water produced by the reaction is also in gas phase, and will be condensed and at that time we recover sensible heat and latent heat of water, which will be used to evaporate isopropanol and bring it to reaction temperature. So energy consumption is minimized with good heat recovery. Of course this is assuming good energy recovery, and technologies similar to heat pumps, using the process stream as fluid, would be appropriate here. Several sentences have been added to explain this thermodynamic aspect of the process, but we don't think that a figure would add more value.

3. Note that these reactions essentially require experiments to be performed at high temperatures, which can make the experimental process potentially dangerous. Therefore, a more detailed experimental procedure needs to be provided for such high temperature experiments.

The paper is not describing experiments, but we included a few warnings, on safety issues - especially on risk of peroxide formation in isopropanol and DIPE, which is not sufficiently taken into account in academic laboratories. The reaction is indeed done at at temperature and pressure, but nothing more than a classical hydrogenation reaction. A few lines have been added about that.

4. The authors need to touch up the current version of the conclusion module as needed.

The conclusion has been modified to point to the best conditions and requirements for catalysts.

Reviewer 3 Report

This manuscript reviews the catalytic dehydration of isopropanol to propylene by providing an overview on both the most used catalysts and the exploited industrial processes. 

Although the topic is of interest, particularly in light of the modern supply chain problems, this manuscript needs extensive work before it can be published since there are several issues that must be addressed as listed below:

- Both the abstract and the introduction are too generic and filled with clichés. The relevance as well as the challanges and issues of the propylene production are not clearly pointed out. This prevents the readers to fully understand the importance of this process. 

- Besides the fact that this entire manuscript is based on only 50 works, there is a general lack of referencing in the text. The are entire sections without any reference and this makes very difficult to find works/patents of interest.

- In Table 2 there is a column titled "Active sites re-sponsible for propylene". I do not know how the authors were able to fill that column since the debate on which among LAS and BAS are the actual active sites is still on going. For instance, reference 33 does not report BAS as the active sites but, quoting, "The Brønsted acid sites [...] were more advantageous for the dehydration of IPA to propylene than Lewis acids". This column is hence highly misleading and as such must be removed. 

- I appreciated the thorough description of the processes patented and commonly used by companies. The same cannot be said the catalysts which is very poor. This section must be dramatically improved before publication. 

Author Response

We thank the reviewer for his careful reading of our publication.

"This manuscript reviews the catalytic dehydration of isopropanol to propylene by providing an overview on both the most used catalysts and the exploited industrial processes. 

Although the topic is of interest, particularly in light of the modern supply chain problems, this manuscript needs extensive work before it can be published since there are several issues that must be addressed as listed below:

- Both the abstract and the introduction are too generic and filled with clichés. The relevance as well as the challenges and issues of the propylene production are not clearly pointed out. This prevents the readers to fully understand the importance of this process. "

The abstract has been modified to make it more clear. The isopropanol dehydration does not intend to substitute propylene production by other means, but rather to find a value to excess acetone, and acetone/isopropanol  which are going to come on the market. So the current challenges of propylene production are not all relevant. Rather, the advantages of using isopropanol as feedstock are pertinent and were described in the text.

"- Besides the fact that this entire manuscript is based on only 50 works, there is a general lack of referencing in the text. There are entire sections without any reference and this makes very difficult to find works/patents of interest."

All relevant patents have been listed in the paper. For academic publications review, another paper will be produced, but as pointed the operating conditions are diverging and that’s not appropriate in the same paper. For what is common knowledge, we preferred to give references to pertinent encyclopedia, in order to limit the number of references.

"- In Table 2 there is a column titled "Active sites re-sponsible for propylene". I do not know how the authors were able to fill that column since the debate on which among LAS and BAS are the actual active sites is still on going. For instance, reference 33 does not report BAS as the active sites but, quoting, "The Brønsted acid sites [...] were more advantageous for the dehydration of IPA to propylene than Lewis acids". This column is hence highly misleading and as such must be removed. "

We agree with the reviewer that the active sites responsible for the propylene formation are still under debate. Considering the remark of reviewer, we have modified the title in the table to “Active sites possibly responsible for propylene” and also added a table foot note: “b Active sites that are beneficial for high propylene yield/selectivity using the reported catalysts at different conditions employed. These sites could be specific for each catalyst reported and may vary with catalysts/experimental conditions.”

Also we added following texts in section 3.3 after the table: The active sites mentioned in the table 2 are the possible sites that can be advantageous for the production of propylene. Also, it is evident that these sites can vary with the catalysts and experimental conditions. Moreover, the LAS and BAS reported here for each catalyst are the sites present in the fresh catalysts which are prone to undergo considerable changes during the dehydration reaction at higher temperatures especially in the presence of water. 

The purpose of the table was also to highlight the diverging opinions among scientist for the sites present on a catalyst that hasn't yet seen the reaction, and which will be modified in-situ. More on that part will be included in ournext publication.

"- I appreciated the thorough description of the processes patented and commonly used by companies. The same cannot be said the catalysts which is very poor. This section must be dramatically improved before publication. "

The type of catalysts used in different patents, as well as those used in the academic literature giving high conversions are cited. This review does not concern a deep description of different catalysts but highlights how the process conditions and catalyst properties are linked together. The paper focusses on the Industrial Processes and catalysts described in the patent literature. What could be extracted from patents regarding the catalysts descriptions was extracted: acidity, surface area, porosity, pore volume, diffusion limitations, sulfate and sodium content, steaming resistance, coke resistance... The paper also highlights the divergence between the conditions used in academic laboratories (low isopropanol partial pressures, diluted stream, and so low residence time) and the industrial process to achieve high propylene purity and productivity (high pressure, high temperature, contact time of several seconds....). In our next publication, we will focus on the catalysts working at low pressure (academic conditions).

Round 2

Reviewer 3 Report

The authors replied in a satisfactory way.